# Antioxidant Activities, Anticancer Activity, Physico-Chemistry Characteristics, and Acute Toxicity of Alginate/Lignin Polymer

**DOI:** 10.3390/molecules28135181

**Published:** 2023-07-03

**Authors:** Nguyen Xuan Hoan, Le Thi Hong Anh, Hoang Thai Ha, Dang Xuan Cuong

**Affiliations:** 1Faculty of Biology and Environment, Ho Chi Minh City University of Industry and Trade, 140 Le Trong Tan, Tan Phu District, Ho Chi Minh 70000, Vietnam; hoannx@hufi.edu.vn; 2Faculty of Food Technology, Ho Chi Minh City University of Industry and Trade, 140 Le Trong Tan, Tan Phu District, Ho Chi Minh 70000, Vietnam; anhlth@hufi.edu.vn; 3Innovation and Entrepreneurship Center, Ho Chi Minh City University of Industry and Trade, 140 Le Trong Tan, Tan Phu District, Ho Chi Minh 70000, Vietnam; cuongdx@hufi.edu.vn

**Keywords:** antioxidant, anticancer, glucosidase, physico-chemistry characteristics, acute toxicity

## Abstract

Alginate/lignin is a synthetic polymer rich in biological activity and is of great interest. Alginate is extracted from seaweed and lignin is extracted from corn stalks and leaves. In this paper, antioxidant activities of alginate/lignin were evaluated, such as total antioxidant activity, reducing power activity, DPPH free radical scavenging activity, and α-glucosidase inhibition activity. Anticancer activity was evaluated in three cell lines (Hep G2, MCF-7, and NCI H460) and fibroblast. Physico-chemistry characteristics of alginate/lignin were determined through FTIR, DSC, SEM_EDS, SEM_EDS mapping, XRD, XRF, and ^1^H-NMR. The acute toxicity of alginate/lignin was studied on *Swiss albino* mice. The results demonstrated that alginate/lignin possessed antioxidant activity, such as the total antioxidant activity, and reducing power activity, especially the α-glucosidase inhibition activity, and had no free radical scavenging activity. Alginate/lignin was not typical in cancer cell lines. Alginate/lignin existed in a thermally stable and regular spherical shape in the investigated thermal region. Six metals, three non-metals, and nineteen oxides were detected in alginate/lignin. Some specific functional groups of alginate and lignin did not exist in alginate/lignin crystal. Elements, such as C, O, Na, and S were popular in the alginate/lignin structure. LD_0_ and LD_100_ of alginate/lignin in mice were 3.91 g/kg and 9.77 g/kg, respectively. Alginate/lignin has potential for applications in pharmaceutical materials, functional foods, and supporting diabetes treatment.

## 1. Introduction

Alginate is a bioactive polymer extracted from brown algae (Phaeophyceae), such as *Sargassum*, *Laminaria*, *Tubinaria*, *Macrocystis*, and *Ascophyllum* grown in tropical and subtropical coasts [1,2,3]. Alginates exist in seaweed cell walls and play a remarkable role in other fields, such as food, functional food, and pharmaceutical, as well as in different industries in the economy [4,5]. Alginate is composed of basic units of (1→4) α-L-guluronic acid (G) and (1→4) β-D-mannuronic acid (M) [6,7,8]. The arrangement of M and G in alginate leads to the different structures and activities of alginate, and this depends on the species of seaweed, the season, and the geographical location of the algae. Alginate possesses different bioactivities, such as antioxidant [9,10], anti-tumor, anti-fungal, neuroprotective, anticancer, and immuno-stimulant bioactivities [11,12,13]. In brown algae, known as *Sargassum* in Vietnam, the alginate content is about 16 to 36 (%wt) of dry algae, and the yield of sargassum seaweed is about 10,000 kg dry per year.

Lignins belong to the polyphenol group in trees, crops, and plants [14]. They are diverse in biological activities, and they are non-toxic. Different plants have different lignins in terms of structure, composition, and bioactivity [15]. Lignins exhibit great potential for applications in numerous areas, such as food, functional food, pharmaceutical, fertilizer, fodder, and fuel. Lignins are found in a mixture of cellulose, hemicellulose, lignin, and extractives of plants [15,16,17]. Softwood, herbaceous plants, and hardwood species include 27–32%, 0–40%, and 21–31% of lignin, respectively [18]. Lignins have structure types known as sulfur lignin (kraft and lignosulfonates lignin) and sulfur-free lignin (alkaline and organosolv lignin) [16,17]. Phenolic hydroxyl and sulfur (SO_3_^2−^ and HSO_3_^−^) groups are found in large amounts in kraft and lignosulfonates lignin, respectively. Kraft and lignosulfonates lignin are dissolved in water [18,19,20]. Organosolv is soluble in organic solvents and insoluble in water. The nitrogen and silicate content in alkaline lignin is more than in other lignins [21,22,23]. In Vietnam, corn stalks yield are about 120–135 tons/ha/year, and the corn planting area is about 1.1 million hectares.

Alginate and lignin possess diverse biological activities, such as anticancer, antibacterial, and antioxidant activities. Physico-chemistry properties of alginate and lignin are interesting. Hence, applications of alginate and lignin in different fields are very diverse; for example, applications include carriers of biologically active ingredients in pharmaceuticals, pharmaceutical materials, food packaging films, drug packaging films, and state-forming agents [2,4,15,16,17]. The combination of alginate and lignin for forming different biomaterials is of great interest, such as in alginate/lignin films. The composite of lignin nanoparticles and sodium alginate removed methylene blue (97.1%) from the aqueous solution, compared to sodium alginate/bulk kraft lignin (82.9%) and pure sodium alginate (77.4%), respectively [24]. The calcium alginate/lignin composite is easy to synthesize, has a low cost, high adsorption efficiencies, good reusability, and perspectives for La(III), Ce(III), Pr(III), and Nd(III) recovery [25]. The alginate/lignin/DAP help the slow-release behavior, which exceeds one month, compared to the uncoated DAP [26]. Survival rate of alginate/lignin-coated Rhizobium cells increased to 12% when they were dry [27]. The alginate/lignin film improved ultraviolet-induced lipid oxidation, was antimicrobial, demonstrated potential in food applications, and had a low cytotoxicity [28]. It can also achieve complete UV resistance and improves mechanical and water-holding properties [29]. Alginate/lignin aerogels increase features of good cell adhesion, and they are non-cytotoxic and suitable for wide applications in tissue engineering and regenerative medicine [30]. However, studies on alginate and lignin particles and properties are lacking, despite the obvious existing dangers of aging, cancer, the lack of medicinal herbs, food, drugs, and functional foods, and environmental pollution to humans.

Therefore, this study focused on the antioxidant activity, anticancer activity, physico-chemistry properties, and acute oral toxicity of alginate/lignin polymer. Lignin and alginate were extracted from the by-products of maize harvest (stalks and leaves) and brown algae processing, respectively.

## 2. Results

### 2.1. Antioxidant Activities

The total antioxidant activity and reducing power activity of alginate/lignin were 218.73 ± 10.45 mg ascorbic acid equivalent/g DW and 479.62 ± 23.18 mg FeSO_4_ equivalent/g DW, respectively (Table 1).

DPPH free radical scavenging capacity of alginate/lignin achieved the highest value of 19.75 ± 4.07 (%) and the lowest value of 0.53 ± 0.78 (%), corresponding to the concentration of 2000 (µg alginate/lignin/mL) and 31.3 (µg alginate/lignin/mL), respectively. A positive control (Trolox) possessed the highest value (81.48 ± 3.30%) and the lowest value (5.20 ± 0.75%) of DPPH free radical scavenging activity, corresponding to 180 (µg Trolox/mL) and 11.25 (µg Trolox/mL), respectively (Table 2). Alginate/lignin and Trolox yielded the average value of alginate/lignin corresponding to 8.39 ± 1.56 (%) and 37.98 ± 1.40 (%), respectively.

α-glucosidase inhibition activity of alginate/lignin was in the range of 27.33 ± 3.62 to 87.62 ± 1.13 (%) with an IC50 of 50.56 ± 0.8 (µg/mL). The lowest and highest value of α-glucosidase inhibition activity corresponded to 27.33 ± 3.62 and 87.62 ± 1.13 (%), respectively, occurring as alginate/lignin content of 15.63 and 250 (µg alginate/lignin/mL), respectively. A positive control of acarbose yielded α-glucosidase inhibition activity of 65.95 ± 0.72 (%), which corresponded to 1000 (µg acarbose/mL) (Table 3).

### 2.2. Anticancer Activities

Alginate/lignin did not exhibit anticancer activity on the NCI H460 line. Camptothecin possessed anticancer activity on four cell lines composed of NCl-H460, fibroblast, hepG2, and MCF-7, corresponding to 64.93 ± 1.58, 47.89 ± 2.58, 57.62 ± 2.06, and 53.89 ± 3.28 (%), respectively, at a camptothecin concentration of 0.007, 2.5, 0.07, and 0.05 (µg camptothecin/mL), respectively. Alginate/lignin showed anticancer activity on two cell lines of fibroblast (14.80 ± 2.00, %) and HepG2 (3.71 ± 4.35%) at the concentration of 1000 (µg alginate/lignin/mL) (Table 4).

### 2.3. Physico-Chemistry Characteristics

FTIR spectrum of alginate/lignin exhibited ten peaks composed of 1610.56, 1417.68, 1041.56, 877.61, 837.11, 623.01, 435.91, 418.55, and 406.98 cm^−1^, corresponding to the functional groups of C=C, O-H, S=O, C-H, C-S, and oxide groups (Figure 1). Alginate/lignin was crystalized at about 100 ^o^C and melted for continuous degradation at 151.5 °C. The region of melting temperature (T_m_) of alginate/lignin was in the range of 148.8 to 159.9 °C with an area of—276.3 J/g (Figure 2). The structure of alginate/lignin did not exist in linear or branched chain but crystallographic characteristics (Figure 3a). Alginate/lignin was found in the shape of spherical particles and was not broken (Figure 4a). The chemical composition of alginate/lignin focused on the range of 1 to 3 and 21–22 keV (Figure 3b) when analysis of them was via XRF spectrum. CK_a_, OK_a_, NaK_a_, SK_a_, and SK_b_ of alginate/lignin occurred in the energy range under 03 keV (Figure 4b). The results of SEM_EDS mapping presented the mass percentage of elements C, O, Na, and S of alginate/lignin corresponding to 22.16 ± 0.08, 41.55 ± 0.13, 27.62 ± 0.12, 8.66 ± 0.08 (%), respectively. The atom percentage of elements C, O, Na, and S of alginate/lignin were 31.2 ± 0.11, 43.91 ± 0.13, 20.32 ± 0.09, and 4.57 ± 0.04 (%), respectively (Appendix A).

Six metals were detected in alginate/lignin composed of Sr, Ta, Fe, Mg, Na, and K. Three non-metals in alginate/lignin were P, Si, and S. Content of six metals achieved a value from 0.1 to 54.9%, corresponding to Ta and Na, respectively. The content of six metals was 1 to 38.7%, corresponding to P and S, respectively. Nineteen oxides of alginate/lignin were shown including to SiO_2_, P_2_O_5_, SO_3_, K_2_O, Fe_2_O_3_, CoO, NiO, ZnO, As_2_O_3_, SrO, Sb_2_O_3_, CdO, SnO_2_, HfO_2_, Ta_2_O_5_, WO_3_, and PbO. The oxide of SO_3_ reached the highest value (43.74%) in the detected oxides of alginate/lignin. CoO, NiO, ZnO, As_2_O_3_, Sb_2_O_3_, and SnO_2_ achieved the lowest value (0.01%) in the oxide that was determined in alginate/lignin (Table 5).

The ^1^H-NMR spectrum of alginate/lignin gave signals at 0.852, 1.068, 1.158, 1.238, 1.361, and 1.535 ppm (Figure 5) with DMSO solvent for running spectrum. The solvent peak was in the signal at 2.5 ppm (Figure 5).

### 2.4. Acute Toxicity

Acute oral toxicity of alginate/lignin material in mice showed that LD_0_ and LD_100_ were 3.91 g/kg and 9.77 g/kg, respectively.
y = 3.6118x^2^ − 31.536x + 67.966; R^2^ = 0.9694.(1)

From Figure 6 and Equation (1), LD_16_ (6.67 g/kg) and LD_84_ (8.91 g/kg) were determined.
(2)$$\mathrm{Normal}\;\mathrm{distribution}:\;S=\frac{{\mathrm{LD}}_{84}-{\;\mathrm{LD}}_{16}}{2}=\frac{8.91-6.67}{2}=1.12.$$

The standard error of LD_50_ was calculated using the following formula:(3)$${\mathrm{SE}}_{\mathrm{LD}50}=\sqrt[{2}]{\frac{k\;\times \;S\;\times \;d}{n}}=\sqrt[{2}]{\frac{0.66\;\times \;1.12\;\times \;0.98}{7}}=0.32.$$ In where, k = 0.66: constant Behrens; S: normal distribution; d: dose jump between two doses LD_50_, d = D_2_ − D_1_ = 8.79 − 7.81 = 0.98 g/kg; and n: average number of tested mice in two batches near the dose LD_50_, n = $\frac{6+8}{2}$ = 7 mice.

Hence, LD_50_ of material alginate/lignin = 8.15 ± 0.32 (g/kg).

The cumulative mortality percentage (%) changed according to the non-linear equation of level 2 (Equation (1)) (Figure 6).

## 3. Discussion

The total antioxidant activity of alginate/lignin was higher than alginate in the previous notice [9,10] and polyphenol of sweet rowanberry cultivars [31]. Sodium alginate achieved a total antioxidant activity of 23.62 ± 17.52 and 188.54 mg ascorbic acid equivalent/g DW, corresponding to brown algae *Sargassum duplicatum* [9] and *Sargassum polycystum* [10], respectively. The reducing power activity of alginate/lignin was higher than that of alginate of brow algae *Cystoseira schiffneri* [32]. The total antioxidant activity of alginate/lignin demonstrated that alginate/lignin possessed antioxidants (218.73 ± 10.45 mg ascorbic acid equivalent/g DW), and that the assay is a pioneer test for further antioxidant assay. The total antioxidant activity of alginate extracted from brown algae *Sargassum polycytum* corresponded to 188.54 mg ascorbic acid equivalent/g DW [10]. Thus, the particles of alginate/lignin could lead to a synergism. Alginates play a role in the absorbent substance for metal, so their reducing power activity is usually high. Alginate/lignin showed the capacity of methylene blue movement [33] and metal in the industry [34]. Reducing power activity is necessary for numerous applications of alginate/lignin. The current study showed that the DPPH free radical scavenging activity of alginate/lignin was higher than one in the notice (EC_50_ = 1.15 mg/mL) [35], and its highest value only corresponded to Trolox at 22.5 µg/mL (Table 2). DPPH free radical scavenging efficiency of alginate/lignin is only 1% of that of Trolox. α-glucosidase inhibition activity of alginate/lignin was ten times more than acarbose. The results demonstrated the potential of alginate/lignin in the application of diabetes medicine, functional food, or pharmaceutical raw materials. The results of α-glucosidase inhibition activity in the current study was higher than previous notices on benzoylphloroglucinols from *Garcinia schomburgakiana* [36] and flavonoids [37]. The former publications did not indicate any antioxidant activities of alginate/lignin.

The results on the anticancer activity of alginate/lignin showed that alginate/lignin was non-selective on tested cancer cells because the inhibition percentage of cancer cells is lower than that of fibroblast cells. Alginate/lignin was not toxic to cancer cells. Alginate/lignin did not have potential in the research and development of K treatment products. The results showed that alginate/lignin is non-toxic.

Functional groups of alginate and lignin was not found in the structure of alginate/lignin. Alginate/lignin only contained functional groups belonging to peaks of 1610.56 to 406.98 cm^−1^. Peaks of 435.91, 418.55, and 406.98 cm^−1^ showed the presence of oxide groups in alginate/lignin. Peak 1417.68 cm^−1^ highlighted the bending of carboxylic acid. The strong bending and stretching of the C=C and C-O groups presented at 837.11 and 1041.56 cm^−1^, respectively [25,38]. The tensile vibration of the group—COO- (1610.56 cm^−1^) and the glycoside ring oscillation (877.61 cm^−1^) exist in the alginate structure. Typical aromatic ring vibration of lignin was at 837.11 cm^−1^. FTIR spectrum of kraft lignin extracted from maize by-products achieved several peaks, such as 3415.52, 1632.16, 1287.60, 1176.76, 1069.08, 1009.52, 880.48, 852.43, and 578.13 (Appendix A). FTIR spectrum did not show the 1738 cm^−1^ peak that exists in the alginate structure (Figure 1). Some typical peaks of lignin disappeared, such as the typical band for the symmetric and asymmetrical vibrations of the CH_3_ group at 2920 and 2850 cm^−1^, respectively, and the characterized peak for the C=C oscillation in the aromatic ring at about 1510 and 1460 cm^−1^ [39]. The band’s disappearance showed the interaction of the two materials to form a stable composite system. DSC spectrum of alginate/lignin showed that the melting temperature was similar to lignin in the notice [40,41]. SEM_EDS, SEM_EDS mapping, XRD, and XRF have shown the morphology, elemental distribution, and content of metal and oxide. Some flattened flakes occurred, which may be the materials that were decomposed/torn and not attached to the particles alginate/lignin at the beginning. Synthetic alginate/lignin particles have a perfect spherical structure. The spheres are unequal in size, intertwined, and stick together. The XRD spectrum of the alginate/lignin system shows the formation of a crystalline phase with sharp peaks on the amorphous background. In previous studies, the XRD spectra of alginate and lignin often showed broad peaks of the carbon system, which are typical for the amorphous or poor crystalline structure of the materials [42,43]. Alginate and lignin formed sphere particles with the enhancement of the crystalline phase on an amorphous substrate. These results demonstrate the applicability of alginate/lignin in pharmaceutical materials and functional foods. The structural properties of alginate/lignin are necessarily analyzed via other solvent systems. In this study, ^13^C NMR spectroscopy did not work because a suitable solvent was not found to conduct the ^1^H NMR spectroscopy for liquid samples. The solvents DMSO, D_2_O, methanol, and acetone could be used for performing NMR spectroscopy of the alginate/lignin samples. The NMR spectrum and conditions for alginate/lignin derived from seaweed and corn by-products have not been noticed. The solvent of D_2_O and 01% CD_3_COOD with an internal standard of DSS and water reduction measurement technique was used for the ^13^C NMR spectroscopy of sodium alginate [10] (Appendix A). The ^1^H NMR analysis of maize stalks lignin was conducted with C_2_D_6_OS solvent and tetramethylsilane as a standard for calibrating the peaks shift [44]. According to Mtibe et al., the shift from 1.5 to 2.4 ppm shows the presence of the group of aromatics [44] (Appendix A).

LD_0_ and LD_100_ are the highest and the lowest doses that cause 0% and 100% death of alginate/lignin-used mice, respectively. The mice were tested over three dose levels with six to eight mice (50% male, and 50% female) per dose, and the lethal dose was of 50% (LD_50_), see Table 5. After drinking alginate/lignin for 30–45 min, the activity of the mice decreased, they laid still, and had diarrhea. Some mice had severe diarrhea and weak breathing and died within 2–4 h. The number of mice with diarrhea was proportional to the numbers of mice administered the oral dose (Table 6). The surviving mice recovered, and diarrhea stopped after 8–24 h of drinking the sample. These mice ate bran pellets and drank water normally. Their stool, urine, and weight were not unusual after recovery. Abnormalities in circulatory function, digestive, sensory reflexes, excretory status, and hair of mice were absent. Mice lived within the first 72 h of observation and for 14 days of follow-up. The recovery time was proportional to the test dose level. Dead mice during the observation period were operated on for further studies. For example, macroscopic examination showed that the internal organs and organs (heart, lungs, liver, stomach, and intestines) were not abnormal. The alginate/lignin material exhibited acute oral toxicity in mice with an LD_50_ value of 8.15 ± 0.32 g/kg, corresponding to an oral dose of 55 kg average adult of 36.443 ± 1.43 g/day (the dose conversion factor between adults and mice is 12.3). After taking the alginate/lignin material, some mice showed diarrhea and decreased activity after 30–45 min; no abnormalities in circulation and sensory reflexes were found; then the mouse died within 2–4 h.

## 4. Materials and Methods

### 4.1. Sample Preparation

#### 4.1.1. Extraction of Sodium Alginate

Brown seaweed *Sargassum polycystum* was selected, cleaned, and stored below 10 °C during transportation to the laboratory. Salt and impurities of the seaweed were removed with fresh water in the laboratory and then the seaweed was dried until its moisture content of 19 ± 1%. After that, the seaweed was ground into a fine powder and stored at 10 °C for further studies. Phlorotannin was separated from brown algae using 96% ethanol in the ratio of 1/10 (*w*/*v*). Fucoidan was separated via H_2_SO_4_ solution (pH 2) for two hours at 80 °C. After fucoidan extraction, the liquid-separated residue was kept for four hours at 60 °C with the ratio Na_2_CO_3_ (pH 9)/dried seaweed (40/1, *v*/*w*) for extracting alginate. Alginate was extracted from the residue. After that, the mixture was hot-filtered to collect the filtrate and added to 10% CaCl_2_ solution (the ratio of CaCl_2_ to alginate was 2.0/1.0) to form a calcium alginate precipitate. Calcium alginate was washed with distilled water and decolorized with 20–30 mL of chlorine solution (1% chlorine solution/100 g calcium alginate antioxidant) for 30 min, and the residual chlorine was removed with fresh water. Bleached calcium alginate was converted to alginic acid by the solution (pH 2.0). Next, the alginic acid was converted to sodium alginate by dissolving the alginic acid in a Na_2_CO_3_ solution with a Na_2_CO_3_/alginic acid ratio of 0.35/1 (*w*/*w*). The mixture was filtered to remove the insoluble fraction of Na_2_CO_3_ solution and then sodium alginate was precipitated in 40% ethanol. The sodium alginate precipitate was filtered and dried at 50 °C under a vacuum to obtain the sodium alginate dry powder.

#### 4.1.2. Extraction of Alkaline Lignin

Corn by-products (corn stalks and leaves) were extracted from polyphenols and chlorophyll by 96% ethanol and were dried in the aerated shade until the moisture was below 20 ± 1% for ground and were stored in breathable solid bags for further studies. Samples were soaked in 4N NaOH at 80 °C for 150 min and filtered. The filtrate was adjusted to pH 5 to form a cellulose precipitate, and then the filtrate was collected. The filtrate was adjusted to 70% ethanol concentration using 96% ethanol to obtain a hemicellulose precipitate. After the removal of hemicellulose, the filtrate was adjusted to pH 2 to collect the precipitated lignin. After centrifugation, the lignin precipitate was washed with clean water and dried at 50 °C under vacuum conditions to constant weight.

#### 4.1.3. Preparation of Alginate/Lignin Particles

Alginate/lignin powder was prepared via a spray-drying method. Sodium alginate in Section 4.1.2 (5% *w*/*v*) was dissolved in de-ionized water at 80 °C for 30 min with stirring at 500 rpm, followed by lignin addition in Section 4.1.3 (2% *w*/*v*, in 0.1 N NaOH) according to the alginate-to-lignin ratio of 90:10 and assimilated at 500 rpm. Alginate/lignin (90/10) spray drying resulted in the best recovery rate, compared with other spray drying conditions. Tween 80 and tripolyphosphate were, in turn, added to the ultrasound-assisted mixture at the rate of 0.1%, compared with the total content of alginate and lignin. The mixture was continuously homogenized during spray drying. The spray drying conditions were as follows: a pump speed of 1500 mL/h, pressure in the chamber of 0.2 atm, air heating temperature of 180 °C, and outlet temperature of 90 °C. Alginate/lignin powder was sifted and preserved in a sealed aluminum bag for further evaluation of the antioxidant activity, anticancer activity, physico-chemistry characteristics, and acute toxicity.

### 4.2. Determination of Antioxidant Activity

#### 4.2.1. Total Antioxidant Activity

Briefly, 100 µL sample was diluted ten times with distilled water and mixed into solution A (0.6 M H_2_SO_4_, 28 mM sodium phosphate, and 4 mM ammonium molybdate) for 5 min. Then, the mixture was incubated at 95 °C for 90 min, and the mixture absorbance at the wavelength of 695 nm was measured. Ascorbic acid was the standard substance [45].

#### 4.2.2. Reducing Power Activity

The determination of reducing power activity was according to the description of Zhu et al., (2002) [46]. Briefly, 500 μL sample was mixed in 0.5 mL of phosphate buffer (pH 7.2), 0.2 mL of 1% K_3_[Fe(CN)_6_], and kept for 20 min at 50 °C. Then, the mixture was added to 500 μL of 10% CCl_3_COOH, 300 μL distilled water, and 80 μL of 0.1% FeCl_3_ for vortexing for 5 min. The absorbance measurement of the compound was at 655 nm with FeSO_4_ used as the standard.

#### 4.2.3. DPPH Free Radical Scavenging Activity

DPPH in 80% methanol was diluted to form 150 µM of DPPH solution and was used immediately. Each well on a 96-well plate was then added to 200 µL of 150 µM DPPH and 25 µL of the sample at different concentrations. The mixture measured the optical density (OD) at 517 nm for 30 min with a jump of 5 min/time. The positive control was Trolox [47].

The percentage of DPPH free radical scavenging activity was calculated using the following formula:(4)$$\mathrm{SC}\%=\left[1-\frac{{\mathrm{OD}}_{t}}{{\mathrm{OD}}_{c}}\right]\times 100\;\left(\%\right),$$

OD_t_ and OD_c_ were the optical density of the test sample and the control, respectively, and these OD values have been subtracted from the OD value of solution wells of non-containing DPPH. SC_50_ value (test concentration of 50% DPPH free radical scavenger) was determined based on a standard curve of optical density values of samples at different concentrations (using Prism software with multi-parameter nonlinear regression and R_2_ > 0.9).

#### 4.2.4. α-Glucosidase Inhibition Activity

Add 120 µL of sample and 20 µL of α-glucosidase (1 unit/mL) to each well on a 96-well plate to incubate the mixture at 37 °C for 15 min. Then, add 20 µL of 5 mM *p*-nitrophenyl-α-D glucopyranoside solution/well and keep for 15 min at 37 °C. Finish the reaction by adding 80 µL of 0.2 M Na_2_CO_3_ solution/well. The absorbance measurement of the mixture was at the wavelength of 405 nm, and the positive control was acarbose [48].

The percentage of α-glucosidase inhibition was calculated using the formula:(5)$$I\%=\left[1-\frac{{\mathrm{OD}}_{t}}{{\mathrm{OD}}_{c}}\right]\times 100\left(\%\right)$$

OD_t_ and OD_c_ were the optical density of the test and control samples, respectively. The OD value of these samples did not include blank OD value (without α-glucosidase). IC50 value (concentration of 50% α-glucosidase inhibitor test substance) was determined based on the standard curve of optical density values of samples at different concentrations (using Prism software with R^2^ > 0.9).

### 4.3. Determination of Anticancer Activity

#### 4.3.1. Cell Culture

The cell lines of breast cancer (MCF-7), lung cancer (NCI H460), liver cancer (Hep G2), and fibroblast was provided by ATCC (Manassas, VA, USA), cultured in E’MEM medium (MCF-7, NCI H460, fibroblast, and Hep G2) supplemented with L-glutamine (2 mM), HEPES (20 mM), amphotericin B (0.025 μg/mL), penicillin G (100 UI/mL), streptomycin (100 μg/mL), and 10% (*v*/*v*) serum FBS bovine fetus and incubated at 37 °C, 5% CO_2_.

#### 4.3.2. Investigation of Cancer Cytotoxic Activity Using the SRB Method

Inoculation of single cells was conducted on 96-well plates at a density of 10^4^ cells/well (for HeLa, Hep G2, fibroblast, and MCF-7 cell lines), 7.5 × 10^3^ cells/well (for the NCI H460 line), and 5 × 10^4^ cells/well for the Jurkat cell line. After 24 h of culture, the cell population was incubated with the probe at different concentrations for 48 h. Then, the total protein fixing of test cells was conducted with a cold solution of 50% trichloroacetic acid (Sigma, St. Louis, MO, USA) (Jurkat alone was 70%) and stained with 0.2% sulforhodamine B solution (Sigma). The results were read with an ELISA reader at two wavelengths of 492 nm and 620 nm. The experiments were triplicated and the results are presented as mean ± standard deviation.

After obtaining the optical density values at 492 nm and 620 nm (denoted OD_492_ and OD_620_):Calculate the value of OD_TS_ = OD_492_ − OD_620_,(6)
Calculate OD_492_ (or OD_620_) = OD_av_ − OD_blank_,(7)

Calculate the percentage (%) of cytotoxicity according to the following formula:(8)$$\%I=\left[1-\frac{{\mathrm{OD}}_{\mathrm{ts}}}{{\mathrm{OD}}_{C}}\right]\times 100\%$$
where,

OD_av_: average OD value of the cells well,OD_blank_: OD value of blank well (no cells),OD_TS_: the OD value of the test sample calculated from Formulas (1) and (2),OD_C_: OD value of the control specimen calculated from Formulas (1) and (2).IC_50_ was determined using Prism software with regression multi-parameter non-linearity and R^2^ > 0.9 [49,50].

### 4.4. Determination of Physico-Chemistry Characteristics

#### 4.4.1. Functional Groups

The sample was measured at a wavelength range from 4000 to 500 cm^−1^ on an IRAffinity-1S of Shimadzu.

#### 4.4.2. Surface Morphology and Elemental Composition

Surface morphology and elemental composition of alginate/lignin were measured via scanning electron microscopy (SEM) and energy dispersive X-ray spectroscopy (EDS) (SEM_EDS), respectively.

#### 4.4.3. Crystallographic Characteristics

Crystallographic characteristics of alginate/lignin were determined according to the X-ray diffraction (XRD) method on a Brucker D2 Phaser instrument. Sample measurement parameters: voltage: 30 KV; current: 10 mA; tube: Cu tube with 1.54184 [Å]; detector: Lynxeye (1D mode); 2ϕ angle: 5–80 degrees; and stepsize (step measurement): 0.02 degrees.

The elemental compositions of alginate/lignin were analyzed via X-ray fluorescence (XRF) method on a Brucker S2 PUMA instrument. Sample measurement parameters: voltage: 20 kV; current: 100 µA, 2000 mW; detector: X-Flash Energy; and energy: −958 eV to 40,138 eV.

#### 4.4.4. Thermal Analysis

Thermal analysis of alginate/lignin was conducted using the differential scanning calorimetry (DSC) method on a NETZSCH DSC 204F1 Phoenix. The surveyed temperature was 30 °C/10.0 (K/min)/400 °C and the atmosphere of N_2_ was 40.0 mL/min/N_2_, 60.0 mL/min.

#### 4.4.5. NMR Spectra

NMR spectra were measured on a Bruker AVANCE Neo 600 MHz instrument at 70 °C, using DMSO as solvent and DSS as an internal standard with a water-reduced measurement technique.

### 4.5. Determination of Acute Toxicity

#### 4.5.1. Test Mice

ICR strain white mice (*Swiss albino*) composed of male and female at six weeks old with a weight of 20 to 26 g, were provided by Nha Trang Institute of Vaccines and Medical Biologicals. Mice were healthy without any abnormal expression in the standard experimental condition for five days. Mice in the cages of size 25 × 35 × 15 cm were supplied with adequate food and water.

#### 4.5.2. Investigation of Acute Oral Toxicity

Ten mice were kept in fasting for at least 12 h before giving them the maximum possible oral dose of the test sample with a volume of 50 mL/kg [51]. Mice were kept in the same condition, fed the same volume of the test sample, and their general movement, behavior, hair state, eating, urination, and death within 72 h were recorded. If the mouse shows no abnormality or dies after 72 h, continue monitoring for 14 days. There are three possible cases:-Case 1: After the mice drank the test sample, the mice did not die, continue determining the highest possible dose of the test sample through the needle without causing the mouse to die (D_max_).-Case 2: After giving the test sample to mice, the mortality rate is 100%, then try ½ dose of the first dose until a minimum amount is lethal to 100% of mice (LD_100_) and a maximum amount that is not lethal to rats (LD_0_). Conduct testing to determine LD_50_.Divide mice into four lots, each batch of 6 mice. Divide the four doses by an exponential interval from LD_0_ to LD_100_. At doses close to LD_50_, the number of mice was increased for more accurate measurements and monitored for 72 h to record the movements of mice, and the number of dead mice in each batch, and to calculate the mortality fraction to find LD_50_.-Case 3: After giving the test sample, the death rate is lower than 100%, the dose of LD_100_ cannot be determined, and the LD_50_ cannot be determined. In this case, it is only possible to determine the maximum dose which is not lethal to the mice, called sub-lethal dose (LD_0_).The LD_50_ value determined via the Behrens method based on two doses close to the LD_50_ lethal dose:

(9)$${\mathrm{LD}}_{50}=D_{1}+\frac{\left(50-\;a\right)\times \;d}{b\;-\;a},$$
in which:
D_1_ is the lethal dose a% of test animals (dose close to 50%).D_2_ is the deadly dose b% of test animals (the upper dose is nearly 50%).d = D_2_ − D_1_ is the dose step between 2 doses near LD_50_.

### 4.6. Data Analysis

Each experiment was triplicated (n = 3) and the results were represented as mean ± SD. ANOVA and statistical analyses were conducted on the software MS Excel 2010. IC50 determination of cell toxicity activity, DPPH free radical scavenging activity, and α-glucosidase inhibition activity were analyzed using GraphPad 7.0 Prism software.

## 5. Conclusions

Alginate/lignin is a potential antioxidant material for pharmaceutical materials, functional foods, and for supporting diabetes treatment. Alginate/lignin exhibited high antioxidant as evident from its total antioxidant activity and reducing power activity, especially its α-glucosidase inhibition activity. Alginate/lignin showed insignificant DPPH free radical scavenging activity, and as well insignificant effect in three cancer cell lines (Hep G2, MCF-7, and NCI H460) and fibroblast cells. Alginate/lignin was in a thermally stable regular spherical shape containing six metals, three non-metals, and nineteen oxides. Typical elements of the alginate/lignin structure were C, O, Na, and S. When alginate was combined with lignin to form a unified crystal, some functional groups of alginate and lignin did not exist in the new alginate/lignin structure. The acute toxicity test of alginate/lignin in mice presented an LD_0_ of (3.91 g/kg) and LD_100_ of (9.77 g/kg). The current study showed potential applications of active alginate/lignin formed by combining alginate from *Sargassum polycystum* and lignin from corn by-products in treating anti-aging, diabetes, and digestive system diseases.

## Figures and Tables

**Figure 1 molecules-28-05181-f001:**
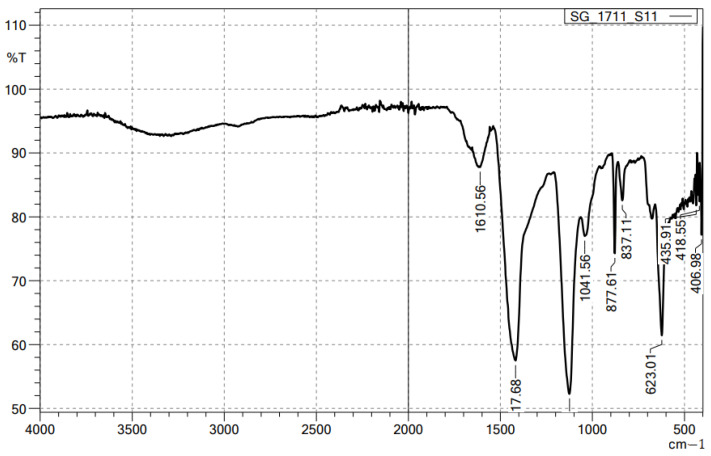
FTIR spectrum of alginate/lignin.

**Figure 2 molecules-28-05181-f002:**
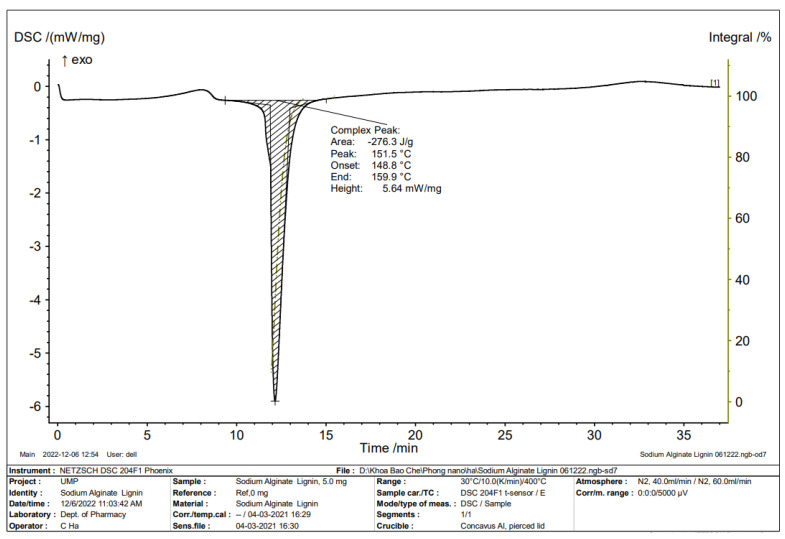
DSC spectrum of alginate/lignin.

**Figure 3 molecules-28-05181-f003:**
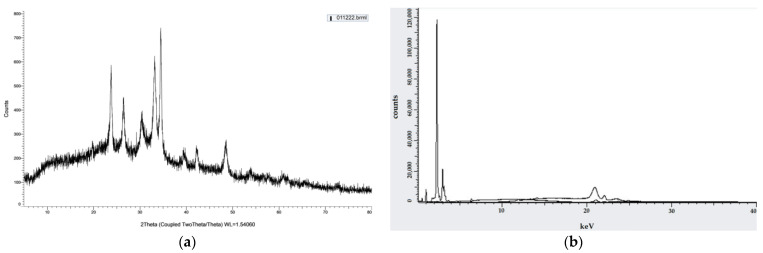
Crystallographic characteristics of alginate/lignin: (**a**) XRD spectrum; (**b**) XRF spectrum.

**Figure 4 molecules-28-05181-f004:**
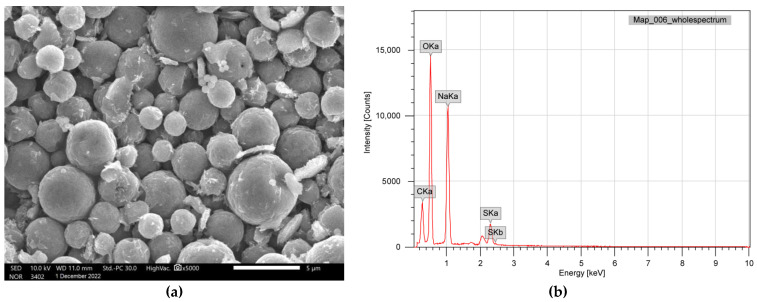
Surface morphology and elemental composition of alginate/lignin: (**a**) SEM_EDS; (**b**) SEM_EDS mapping.

**Figure 5 molecules-28-05181-f005:**
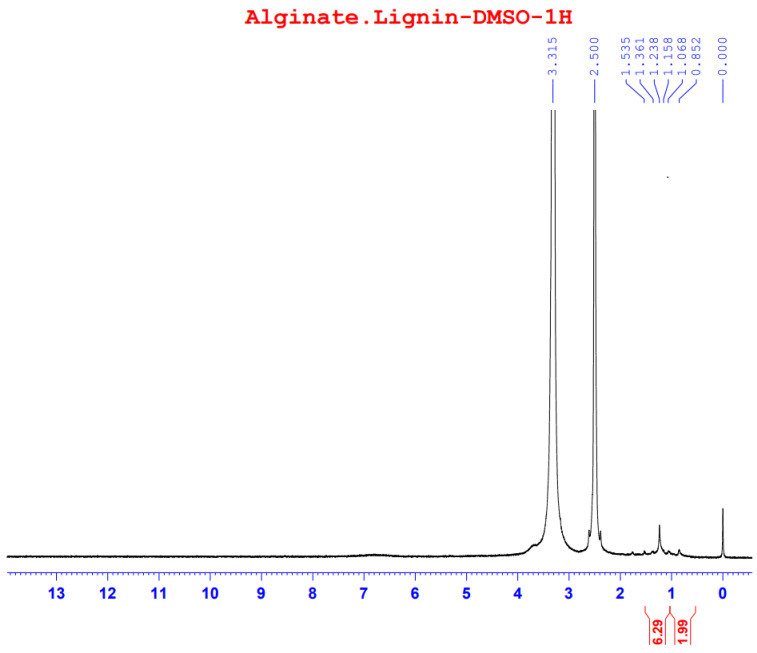
^1^H spectrum of alginate/lignin.

**Figure 6 molecules-28-05181-f006:**
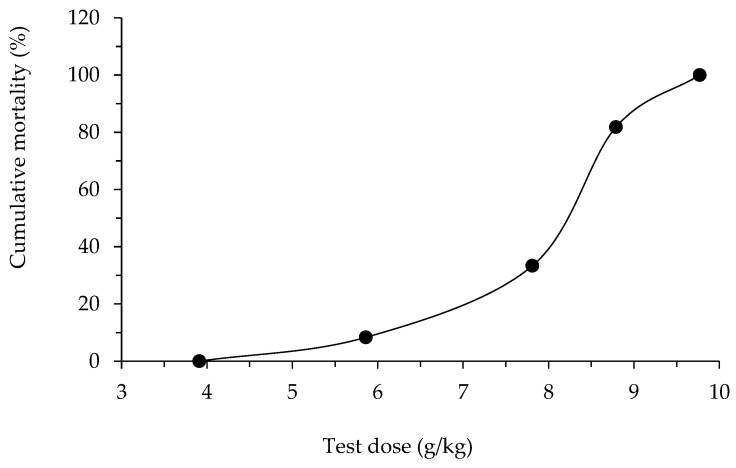
Cumulative mortality by oral dose of alginate/lignin.

**Table 1 molecules-28-05181-t001:** Total antioxidant activity and reducing power activity of alginate/lignin.

Sample	Total Antioxidant Activity (mg Ascorbic Acid Equivalent/g DW)	Reducing Power Activity (mg FeSO_4_ Equivalent/g DW)
Alginate/lignin	218.73 ± 10.45	479.62 ± 23.18

**Table 2 molecules-28-05181-t002:** DPPH free radical scavenging activity of alginate/lignin and Trolox.

Sample	Concentration (µg/mL)	DPPH Free Radical Scavenging Activity (%)
Alginate/lignin	4000	15.31 ± 1.90
	2000	19.75 ± 4.07
	1000	15.72 ± 1.84
	500	9.59 ± 1.13
	250	4.12 ± 1.03
	125	1.61 ± 0.79
	62.5	0.50 ± 0.96
	31.3	0.53 ± 0.78
Trolox	180	81.48 ± 3.30
	90	52.01 ± 0.44
	45	30.34 ± 1.69
	22.5	20.87 ± 0.83
	11.25	5.20 ± 0.75

**Table 3 molecules-28-05181-t003:** α-glucosidase inhibition activity of alginate/lignin and acarbose.

Sample	Concentration (µg/mL)	α-Glucosidase Inhibition Activity (%)	IC_50_ (µg/mL)
Alginate/lignin	250	87.62 ± 1.13	50.56 ± 0.8
	125	79.74 ± 3.04	
	62.5	49.55 ± 4.83	
	31.25	32.96 ± 5.07	
	15.63	27.33 ± 3.62	
Acarbose	1000	65.95 ± 0.72	

**Table 4 molecules-28-05181-t004:** Anticancer activities of alginate/lignin.

Cancer Cell Lines	Sample	Concentration (µg/mL)	Cell Toxicity (%)
NCl-H460	Alginate/lignin	1000	−5.66 ± 6.33
	H_2_O	10%	3.34 ± 7.05
	Camptothecin	0.007	64.93 ± 1.58
Fibroblast	Alginate/lignin	1000	14.80 ± 2.00
	H_2_O	10%	6.10 ± 3.24
	Camptothecin	2.5	47.89 ± 2.58
HepG2	Alginate/lignin	1000	3.71 ± 4.35
	H_2_O	10%	−5.73 ± 4.70
	Camptothecin	0.07	57.62 ± 2.06
MCF-7	Alginate/lignin	1000	1.76 ± 9.14
	H_2_O	10%	−5.57 ± 3.95
	Camptothecin	0.05	53.89 ± 3.28

**Table 5 molecules-28-05181-t005:** Composition and content of metal and oxide in alginate/lignin.

Metal and Non-Metal of Alginate/Lignin	Content (%)	Oxide of Alginate/Lignin	Content (%)	Oxide of Alginate/Lignin	Content (%)
Sr	0.10	SiO_2_	2.10	SrO	0.03
Ta	0.10	P_2_O_5_	1.23	Sb_2_O_3_	0.01
Fe	0.20	SO_3_	43.74	CdO	0.02
Mg	0.70	K_2_O	0.21	SnO_2_	0.01
Na	54.90	Fe_2_O_3_	0.12	HfO_2_	0.02
K	0.50	CoO	0.01	Ta_2_O_5_	0.03
P	1.00	NiO	0.01	WO_3_	0.03
Si	3.50	ZnO	0.01	PbO	0.02
S	38.7	As_2_O_3_	0.01	Bi_2_O_3_	0.02
				Na_2_O	51.8

**Table 6 molecules-28-05181-t006:** Number of dead/alive mice in each batch of alginate/lignin.

Dosage (g/kg)	Actual Number			Cumulative Amount				Mice Have Diarrhea	
	Deaths	Lives	Total	Deaths	Lives	Total	% Deaths	Quantity	Recovery Time (h)
3.91	0	6	6	0	17	17	0.00	2	8–10
5.86	1	5	6	1	11	12	8.33	3	12–15
7.81	2	4	6	3	6	9	33.33	4	12–18
8.79	6	2	8	9	2	11	81.82	8	12–24
9.77	6	0	6	15	0	15	100.00	6	Dead mouse

## Data Availability

Not applicable.

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
