# Peer review of "Antioxidant Activities, Anticancer Activity, Physico-Chemistry Characteristics, and Acute Toxicity of Alginate/Lignin Polymer"

_molecules, 2023, doi:10.3390/molecules28135181_

Round 1
Reviewer 1 Report
I consider that the document contains information that does not contribute new information and that it is necessary to optimize and correct it. As well as include data that is discussed but does not appear in the document.
Table 4. what is the metal R/R0?
Si and P are not metal
Page 7 line 155
For a better understanding it is recommended to indicate the values ​​that are being compared in discussion section:
-The total antioxidant activity of alginate/lignin was higher than one of alginate in the previous notice (indicate the value) DPPH free radical scavenging activity of alginate/lignin in the current study was higher than one in the notice [28] (missing data)
Line 158. The total antioxidant activity of alginate/lignin exhibited alginate/lignin possessing antioxidants, and the assay is a pioneer test for further antioxidant assay. (Missing data)
Page 2 line 72. Results Section
The total antioxidant activity and reducing power activity of alginate/lignin was 218.73 ± 10.45 mg ascorbic acid equivalent/g DW and 479.62 ± 23.18 mg FeSO4 equivalent/g DW. Please show result table for total antioxidant activity and reducing power activity.
Table 3. The activity showed is not anticancer activity is antiproliferative activity, please review the concept.
Section 2.3 Physicochemical characteristics.
What is the information of the spectrum of figure 5 if no signals are observed or it is not discussed in the document. 13NMR data is missing.
Author Response
Dear
The reviewer of Molecules for the manuscript ID 2424230 in Section: Applied Chemistry
The authors sincerely thank the reviewer for your help.
The author repaired according to the commends of the review, specifically as follows:
- Table 4. what is the metal R/R0?: removed
- Si and P are not metal: repaired and added, in line 141 to line 144, Table 5 (added Table 1, so Table 4 into Table 5)
- Page 7 line 155
For a better understanding it is recommended to indicate the values ​​that are being compared in discussion section:
-The total antioxidant activity of alginate/lignin was higher than one of alginate in the previous notice (indicate the value)
Added in lines 177 to 180
DPPH free radical scavenging activity of alginate/lignin in the current study was higher than one in the notice [28] (missing data)
Added in line 191
Line 158. The total antioxidant activity of alginate/lignin exhibited alginate/lignin possessing antioxidants, and the assay is a pioneer test for further antioxidant assay. (Missing data)
Added in lines 182 to 186
Page 2 line 72. Results Section
The total antioxidant activity and reducing power activity of alginate/lignin was 218.73 ± 10.45 mg ascorbic acid equivalent/g DW and 479.62 ± 23.18 mg FeSO4 equivalent/g DW. Please show result table for total antioxidant activity and reducing power activity.
Added Table 1
Table 3. The activity showed is not anticancer activity is antiproliferative activity, please review the concept.
Adjusted in lines 202 and 204.
Section 2.3 Physicochemical characteristics.
What is the information of the spectrum of figure 5 if no signals are observed or it is not discussed in the document. 13NMR data is missing.
Added in lines 213 to 215, lines 235 to 244.
The authors hope to continue to receive the reviewer's help and early reply.
Best regards!
Reviewer 2 Report
The manuscript has interesting subject taking into account that alginate and lignin possess various biological activities and a lot of applications in different fields.
The manuscript is well written and is focused on the extraction of lignin and alginate from biomass and on the antioxidant and anticancer activity and also acute oral toxicity.
To improve the manuscript for consideration for publication I have some suggestions for authors:
1.In the introduction it is necessary to highlight the importance of the combination alginate/ lignin with literature data.
2. For the mixture alginate /lignine biopolimers, physico-chemical characterisation was realized. Please add also the same characteristics ( IR, RMN, etc) also for individual biopolymers, maybe some of them in a supplimentary file.
3. To show why it is more interesting to use this mixture, proposed by the authors, the analysis should be compared with those for pure alginate and lignin. A mixture can lead to a synergism or antagonism, so the antioxidant and antitumoral activity should be presented in comparison with the both extracted compounds, individually.
4. Why did the authors choose this ratio between alginate and lignine for this study? It should be explained.
5. The conclusions must highlight the importance of the results obtained in this study.
Author Response
Dear
Editor in Chief
The reviewer of Molecules for the manuscript ID 2424230 in Section: Applied Chemistry
The authors sincerely thank the reviewer for your help.
The authors repaired according to the commends of the review, specifically as follows:
- In the introduction it is necessary to highlight the importance of the combination alginate/ lignin with literature data.
Added in lines 63 to 66, 67 to 76.
- For the mixture alginate /lignine biopolimers, physico-chemical characterisation was realized. Please add also the same characteristics ( IR, RMN, etc) also for individual biopolymers, maybe some of them in a supplimentary file.
Added in lines 213 to 215, lines 235 to 244, and supplementary Figure S1 to S3
- To show why it is more interesting to use this mixture, proposed by the authors, the analysis should be compared with those for pure alginate and lignin. A mixture can lead to a synergism or antagonism, so the antioxidant and antitumoral activity should be presented in comparison with the both extracted compounds, individually.
In this study, we focused on alginate/lignin, so we did not evaluate alginate and lignin individually. However, we have added an explanatory comparison, specifically: Added in lines 177 to 180, 182 to 186, 191, 202 and 204.
- Why did the authors choose this ratio between alginate and lignine for this study? It should be explained.
Added in lines 301 to 302.
- The conclusions must highlight the importance of the results obtained in this study.
Repaired.
The authors hope to continue to receive the reviewer's help and early reply.
Best regards!
Round 2
Reviewer 2 Report
thanks for the changes made.